# The Role of SHIP1 on Apoptosis and Autophagy in the Adipose Tissue of Obese Mice

**DOI:** 10.3390/ijms21197225

**Published:** 2020-09-30

**Authors:** Jae Hun Jeong, Eun Bee Choi, Hye Min Jang, Yu Jeong Ahn, Hyeong Seok An, Jong Youl Lee, Gyeongah Park, Eun Ae Jeong, Hyun Joo Shin, Jaewoong Lee, Kyung Eun Kim, Gu Seob Roh

**Affiliations:** Department of Anatomy and Convergence Medical Science, Bio Antiaging Medical Research Center, Institute of Health Sciences, College of Medicine, Gyeongsang National University, Jinju, Gyeongnam 52727, Korea; wogns3764@gmail.com (J.H.J.); dmsql2274@naver.com (E.B.C.); gpals759@naver.com (H.M.J.); ahnujung@naver.com (Y.J.A.); gudtjr5287@hanmail.net (H.S.A.); jyv7874v@naver.com (J.Y.L.); imkapark@gmail.com (G.P.); jeasky44@naver.com (E.A.J.); k4900@hanmail.net (H.J.S.); woongs1111@gmail.com (J.L.); kke-jws@hanmail.net (K.E.K.)

**Keywords:** SHIP1, macrophage, apoptosis, autophagy, adipose tissue, obesity

## Abstract

Obesity-induced adipocyte apoptosis promotes inflammation and insulin resistance. Src homology domain-containing inositol 5′-phosphatase 1 (SHIP1) is a key factor of apoptosis and inflammation. However, the role of SHIP1 in obesity-induced adipocyte apoptosis and autophagy is unclear. We found that diet-induced obesity (DIO) mice have significantly greater crown-like structures and terminal deoxynucleotidyl transferase deoxyuridine triphosphate (dUTP) nick-end labeling (TUNEL)-positive cells than ob/ob or control mice. Using RNA sequencing (RNA-seq) analysis, we identified that the apoptosis- and inflammation-related gene Ship1 is upregulated in DIO and ob/ob mice compared with control mice. In particular, DIO mice had more SHIP1-positive macrophages and lysosomal-associated membrane protein 1 (LAMP1) as well as a higher B-cell lymphoma 2 (Bcl-2)-associated X protein (Bax)/Bcl-2 ratio compared with ob/ob or control mice. Furthermore, caloric restriction attenuated adipose tissue inflammation, apoptosis, and autophagy by reversing increases in SHIP1-associated macrophages, Bax/Bcl2-ratio, and autophagy in DIO and ob/ob mice. These results demonstrate that DIO, not ob/ob, aggravates adipocyte inflammation, apoptosis, and autophagy due to differential SHIP1 expression. The evidence of decreased SHIP1-mediated inflammation, apoptosis, and autophagy indicates new therapeutic approaches for obesity-induced chronic inflammatory diseases.

## 1. Introduction

Obesity-induced hypertrophy of adipose tissue causes adipocyte apoptosis, exacerbating inflammation and eventually aggravating metabolic disorders, such as insulin resistance and type 2 diabetes [1]. Macrophages play an important role both in the initiation and resolution of inflammation induced by obesity-induced insulin resistance [2]. In particular, macrophages develop distinct functional phenotypes by undergoing different classical M1 and alternative M2 macrophages [3]. Accumulation of proinflammatory M1 macrophages, for example, was found to be associated with obesity and insulin resistance, while anti-inflammatory M2 macrophages were largely absent in obese adipose tissue in mice [4]. Furthermore, infiltration of M1 macrophages in adipose tissue exacerbates adipocyte apoptosis and autophagy [5,6]. Although regulators of adipose tissue macrophages have been previously assessed [7], the role of macrophages on apoptosis and autophagy in adipose tissue has yet to be fully elucidated.

Src homology domain-containing inositol 5′-phosphatase 1 (SHIP1), an enzyme with phosphatase activity, is predominantly expressed in hematopoietic cells [8]. SHIP1 acts as a negative regulator of survival, proliferation, and autophagy activation through the phosphoinositide 3-kinase (PI3K) pathway by translocating membranes and hydrolyzing the PI3K-generated second messenger, PIP3 to PIP2, after extracellular stimulation [9]. SHIP1^−/−^ peritoneal and alveolar macrophages promote anti-inflammatory cytokines and protect tissues from inflammation due to the immunosuppressive effects of SHIP1 inhibition [10]. Furthermore, SHIP1 regulates phagosome maturation in macrophages by modulating PI3K [11]. However, it remains unclear as to whether SHIP1 expression of adipose tissue macrophages induced by obesity contributes to apoptosis and autophagy.

In this study, we investigated the role of altered SHIP1 on obesity-induced adipose tissue apoptosis and autophagy. We examined the effect of diet-induced obesity (DIO) or leptin deficiency on SHIP1-positive macrophages, apoptosis, and autophagy in mice. Furthermore, we determined the effects of caloric restriction (CR) on SHIP1-associated macrophages; the B-cell lymphoma 2 (Bcl-2)-associated X protein (Bax)/Bcl-2 ratio; and autophagy-related p62, microtubule-associated protein 1A/1B-light chain 3B (LC3B), and lysosomal-associated membrane protein 1 (LAMP1) expression to assess the therapeutic potential of modulating SHIP1 expression to treat obesity-induced metabolic syndrome. To the best of our understanding, this is the first report demonstrating that SHIP1 may play an important role in adipose tissue apoptosis and autophagy in obese mice.

## 2. Results

### 2.1. Macrophage Infiltration and Inflammation in the Adipose Tissue of DIO and ob/ob Mice

DIO and ob/ob mice exhibited greater body weight, fat mass, and epididymal fat pad weight than control (CTL) mice (Figure 1A–D). Serum leptin was significantly higher in DIO mice (86.14 ± 6.63 ng/mL) compared with CTL mice (10.38 ± 6.27 ng/mL) (*p* < 0.0001). Leptin-deficient ob/ob mice do not have any leptin production. Notably, although the adipocyte size was significantly higher in ob/ob mice than in DIO mice, hematoxylin and eosin (H&E) staining revealed more crown-like structures (CLSs) in DIO mice than in CTL or ob/ob mice (Figure 1E–G). Additionally, tumor necrosis factor-α (Tnf-α) and monocyte chemoattractant protein 1 (Mcp1) mRNA expression levels were greater in DIO and ob/ob mice than in CTL mice (Figure 1H), whereas CC chemokine ligand 5 (CCL5) and C-C motif receptor 5 (CCR5) mRNA expression levels were greater in DIO mice than in CTL or ob/ob mice (Figure 1H). However, no significant change was shown in interleukin-10 (Il-10) and arginase 1 (Arg1) mRNA levels (Figure 1I).

### 2.2. Differential Gene Expression in the Adipose Tissue of DIO and ob/ob Mice

Obesity aggravates inflammation and apoptosis in adipose tissue [12,13]. Therefore, we performed next-generation RNA-seq analysis to assess differentially expressed genes related to apoptosis and inflammation in adipose tissue. Principal component analysis identified several differences in gene expression between the three groups (Figure 2A). Compared with CTL mice, DIO and ob/ob mice had 5681 common genes that were differentially expressed (*p* < 0.05), of which 2523 genes displayed a |fold change| < 1.60 (Figure 2B). These differentially expressed genes affect many biological processes, cellular components, and molecular functions. Additionally, several genes may potentially cause inflammation and apoptosis, such as those involved in the regulation of IL-1α secretion, the B cell receptor signaling pathway, and autophagolysosome assembly (Appendix A). Notably, several differences in the expression levels of genes promoting inflammation and apoptosis were identified (Figure 2C,D and Appendix A). Nearly twofold increases in Ship1 and Bax expression were observed in common genes of DIO mice and ob/ob mice relative to those of CTL mice, whereas a significant decrease in Bcl-2 expression was found in common genes of DIO and ob/ob mice compared with those of CTL mice. (Figure 2E–G). These findings indicate that SHIP1 is associated with adipose tissue apoptosis and inflammation.

### 2.3. Effects of DIO and Leptin Deficiency on Inflammation and Apoptosis in Adipose Tissue

Next, we determined whether DIO or ob/ob was associated with changes in SHIP1 expression of hematopoietic cells including macrophages in adipose tissue. Western blot analysis showed that DIO mice and ob/ob mice had higher SHIP1 expression than CTL mice (Figure 3A). Immunohistochemical staining revealed that DIO and ob/ob mice had more SHIP1-positive CD68 macrophages than CTL mice, with DIO mice having the most (Figure 3B). Perilipin-free apoptotic adipocytes surrounded by macrophages play a crucial role in adipose tissue inflammation [14]. To investigate whether SHIP1-positive macrophages affect lipid droplet clearance, we performed double immunofluorescence staining for SHIP1 and perilipin 1. DIO and ob/ob mice exhibited more SHIP1-positive cells around perilipin 1–free adipocytes than CTL mice (Figure 3C). Additionally, double immunofluorescence staining for SHIP1 and terminal deoxynucleotidyl transferase dUTP nick-end labeling (TUNEL) revealed a higher number of SHIP1-positive apoptotic cells in DIO and ob/ob mice than in CTL mice, with the highest in DIO mice (Figure 3D). The Bax/Bcl-2 ratio, the hallmark of cell apoptosis, and cleaved caspase-3 were significantly greater in DIO and ob/ob mice than in CTL mice, with the greatest levels found in DIO mice (Figure 3E and Appendix A). Because of the key role of PI3K in apoptosis, we investigated Akt protein targets of PI3K. DIO and ob/ob mice exhibited significantly lower levels of Akt phosphorylation than CTL mice (Figure 3F,G). Together, these results indicate that increased adipose tissue SHIP1 expression in DIO and ob/ob mice may contribute to adipose tissue inflammation and apoptosis induced by obesity.

### 2.4. Effects of DIO and Leptin Deficiency on Autophagy in Adipose Tissue

Autophagy plays an important role in inflammation by influencing the development of macrophages [15]. Triple immunofluorescence revealed that 4′,6-diamidino-2-phenylindole (DAPI) and LAMP1 were co-localized and significantly more perilipin 1-free adipocytes were surrounded by LAMP1-positive autolysosomes in DIO mice than in ob/ob and CTL mice (Figure 4A). Additionally, DIO mice and ob/ob mice had higher p62 expression than CTL mice (Figure 4B,C). However, ob/ob mice had a greater LC3B II/I ratio than DIO or CTL mice (Figure 4D). Furthermore, DIO mice had significantly higher LAMP1 than ob/ob mice or CTL mice (Figure 4E). Taken together, autophagy activity can be upregulated in adipose tissue of obese mice by influencing inflammation and apoptosis.

### 2.5. Effects of CR on SHIP1 Expression Related to Apoptosis and Autophagy in the Adipose Tissue of DIO and ob/ob Mice

Our previous studies demonstrated that CR attenuates body weight in DIO and ob/ob mice [16,17]. Compared with DIO and ob/ob mice, calorie-restricted DIO and ob/ob mice exhibited fewer SHIP1-positive CD11b, a macrophage marker (Figure 5A and Figure 6A). Additionally, CR significantly reduced SHIP1 expression, Bax/Bcl-2 ratio, and cleaved caspase-3 in the adipose tissues of DIO and ob/ob mice (Figure 5B,C, Figure 6B,C, and Appendix A).

Many LAMP1-positive cells around perilipin 1-free adipocytes in DIO and ob/ob mice were attenuated by CR (Figure 5D and Figure 6D). Furthermore, Western blot analysis showed that CR reduced increases in p62, the LC3BII/I ratio, and LAMP1 expression in the adipose tissues of DIO and ob/ob mice (Figure 5E,F and Figure 6E,F). These findings indicate that CR can attenuate adipose tissue inflammation, apoptosis, and autophagy by modulating SHIP1 protein expression.

## 3. Discussion

Adipose tissue inflammation, causing adipocyte apoptosis, has been reported to occur in response to insulin resistance and metabolic diseases [18]. Various human disorders, along with inflammation, include an irregular expression or deficiency of inositol phosphatases [19]. According to previous research, the inhibitor of SHIP1 increases the immunoregulatory capacity that reverses obesity and metabolic syndrome [20]. To our knowledge, the present study is the first to identify the fact that DIO and ob/ob mice exhibit higher apoptosis levels and autophagy counts related to SHIP1 in adipose tissue than CTL mice. However, CR reduced increased adipose tissue SHIP1 expression in DIO or ob/ob mice. Furthermore, CR mice significantly attenuated Bax/Bcl-2 ratio and LAMP1 in DIO and ob/ob mice. Taken together, our results indicate that SHIP1 in obese mice can play a critical role in adipocyte apoptosis and autophagy.

Leptin is a cytokine and, similar to other cytokines such as interleukin-6 and TNF-α, can induce a pro-inflammatory response via its activation of the PI3K/Akt signaling pathway [21]. Hyperleptinemia occurs in response to a high-fat diet (HFD), causing leptin resistance [22]. We found no significant differences in TNF-α mRNA between DIO mice and ob/ob mice, but DIO mice showed significantly lower PI3K and p-Akt/t-Akt ratio than ob/ob mice. This difference may be explained by differential leptin concentration between DIO (hyperleptinemia) and leptin-deficient ob/ob mice. We found that DIO mice had more CLSs and CD68-positive macrophages in epididymal adipose tissue than ob/ob mice. However, another study demonstrated that increased macrophage infiltration of adipose tissue was positively correlated with fat mass in DIO and ob/ob mice [5]. This difference may be explained by the age of mice in each study—the previous study used 12-week HFD-fed mice with hyperleptinemia, whereas we used 20-week HFD-fed mice. However, 12-week HFD mice display pro-inflammatory properties of the non-leptin-mediated pathway. On the basis of these understandings, we hypothesize that an unknown pathway other than leptin causes adipose tissue inflammation and apoptosis.

SHIP1 is closely associated with inflammation and apoptosis in hematopoietic cells [10,23]. In this study, we found significantly higher SHIP1 protein levels, more SHIP1-positive TUNEL cells, and a greater Bax/Bcl-2 ratio in DIO mice than in ob/ob mice. This differential SHIP1 level may play a crucial role in adipocyte apoptosis between DIO and ob/ob mice. However, contradictory roles for SHIP1 may exist in rodents—one study suggested that the inflammation response is aggravated by SHIP1 deficiency [24], but another identified a bone marrow-derived, macrophage-specific SHIP1 deficiency that reduces the pro-inflammatory response [25]. In this study, upregulated SHIP1 apoptosis in obese mice indicates that obesity-induced SHIP1 aggravates adipose tissue inflammation. Additionally, we found that SHIP1-positive macrophages are reduced in calorie-restricted DIO and ob/ob mice. In other study, insulin resistance in HFD-fed mice increased by bacterial lipopolysaccharide (LPS), and these mice promoted pro-inflammatory cytokine secretion, which then triggered metabolic diseases [26]. Moreover, SHIP1 was upregulated from LPS stimulation in bone marrow-derived macrophages, and SHIP1 overexpression increased LPS-induced TNF-α production [10]. In our study, DIO mice had significantly more SHIP1-positive macrophages than ob/ob mice. These findings indicate that elevated leptin-mediated inflammation of adipose tissue through endotoxin mediates pro-inflammatory properties of SHIP1, which may play a crucial role in adipose tissue inflammation.

In addition to its inflammatory and apoptotic role, it has been reported that SHIP1 negatively regulates PI3K-mediated signaling and hence represses the cell differentiation and survival [27]. Upregulating miRNA-210 in 3T3-L1 cells promoted 3T3-L1 adipogenesis through SHIP1 as a negative regulator of PI3K/Akt signaling [28]. This blocking adipogenesis may be linked with the apoptotic process of adipocytes. Our findings suggest that the activation of SHIP1 in obese adipocytes inhibits the PI3K/Akt pathway and then causes adipocyte apoptosis. On the other hand, perilipin 1 expression is elevated in obese animals and humans [29]. In our study, however, DIO mice exhibited more perilipin 1-free adipocytes than ob/ob mice or CTL mice. These findings are consistent with evidence that dead adipocytes (perilipin 1-free) are surrounded by macrophages [14,30]. It has been suggested that perilipin 1-free staining be used as a marker of adipocyte death. Thus, we suggest that DIO mice have the pronounced apoptosis of adipocyte compared with CTL or ob/ob mice. These data suggest SHIP1 around perilipin 1-free adipocytes can act as a negative regulator of adipogenesis. However, to determine whether the SHIP1 effect is specific to adipogenesis in pre-adipocytes, a future in vitro study is required to investigate SHIP1-specific RNA interference and SHIP1 overexpression in primary adipocytes or macrophages.

Autophagy is a process of lysosomal degradation that is initiated by the emergence of double-membrane vesicles, known as autophagosomes [31]. One study showed that patients with obesity have increased autophagy activity in adipose tissues [32]. Consistent with this understanding, DIO and ob/ob mice had more autophagy protein p62 and greater LC3B II/I ratio than CTL mice. However, ob/ob mice had significantly greater LC3B II/I ratio than in DIO or CTL mice. These findings indicate that leptin treatment inhibits the autophagy of white adipose tissue in ob/ob mice [33]. By contrast, DIO mice unexpectedly had slightly higher LAMP1 around perilipin-free adipocytes such as SHIP1-positive macrophages than ob/ob or CTL mice. However, hypertrophic adipose tissue reportedly induces lysosome biogenesis without inflammatory polarization of adipose tissue macrophages [34]. This difference may be explained by the way in which increased SHIP1-positive macrophages induced by HFD enhance lysosome biogenesis, yet increased leptin level in DIO mice may inhibit autophagosome maturation. Obesity-related inflammation initiates an autophagy–lysosomal response and then causes degradation of perilipin 1 [35]. Consistent with this understanding, we found that the LC3B II/I ratio, p62, and LAMP1 are reduced in calorie-restricted DIO and ob/ob mice. Taken together, this evidence suggests that SHIP1-positive macrophages around perilipin-free adipocytes trigger autophagy–lysosomal response and apoptosis for clearance of lipid cell debris.

In conclusion, these findings indicate that adipocyte inflammation, apoptosis, and autophagy induced by obesity is improved by CR through the SHIP1-mediated pathway. Although the question of whether SHIP1 modulation is deleterious or protective still remains, our findings suggest that SHIP1 upregulation in obese adipose tissue is closely associated with inflammation, apoptosis, and adipocyte remodeling.

## 4. Materials and Methods

### 4.1. Animals

Three-week-old male C57BL/6 mice and ob/ob mice were purchased from Central Laboratory Animal Inc. (Seoul, South Korea). Animal experiments were performed in accordance with the National Institutes of Health guidelines on the use of laboratory animals. The Animal Care Committee for Animal Research at Gyeongsang National University approved the study protocol (GNU-160530-M0025). The mice were individually housed under a 12-h light/dark cycle. Wild-type mice were fed ad libitum with normal standard diet chow (*n* = 10; CTL group) or a HFD (*n* = 10; DIO group, 60% kcal from fat; Research Diet, Inc., New Brunswick, NJ, USA); ob/ob mice were fed ad libitum with normal standard diet chow. To determine the effects of CR on apoptosis and inflammation induced by DIO or ob/ob, we assigned 8-week-old DIO and 10-week-old ob/ob mice to ad libitum or calorie-restricted diets for 12 weeks, as previously described [36,37]. Animals underwent whole-body composition analysis using the EchoMRI (Echo Medical Systems, Houston, TX, USA) to quantify body fat.

### 4.2. Enzyme-Linked Immunosorbent Assay (ELISA)

After overnight fasting, mice were anesthetized with zoletil (5 mg/kg; Virbac Laboratories, Carros, France). Blood samples were extracted from the heart and centrifuged. Serum leptin was measured using mouse ELISA (R&D Systems, Minneapolis, MN, USA), according to the manufacturer’s instructions.

### 4.3. RNA-seq Analysis

C&K Genomics (C&K Genomics Inc., Seoul, Korea) performed RNA-seq analysis from the adipose tissue of CTL, DIO, and ob/ob mice at 25 weeks of age (*n* = 3 mice per group). The sequencing library was constructed using Illumina’s TruSeq RNA Prep kit (illumina Inc., San Diego, CA, USA), and data generation was performed using the NextSeq 500 platform (illumina Inc.) following the manufacturer’s protocol. RNA-seq analysis was performed as previously described [17].

### 4.4. Real-time Reverse Transcriptase Polymerase Chain Reaction

Total RNA from epididymal adipose tissue (*n* = 4–6 per group) was prepared using a previously reported method [17]. PCR primers used for this study are presented in Appendix A. Expression was normalized to the level of glyceraldehyde 3-phosphate dehydrogenase as an internal control.

### 4.5. Histological Analysis

For histological studies, samples of epididymal fat pads (*n* = 3 mice/group) were embedded in paraffin and cut into 5-mm sections. CLSs in adipose tissue were identified using H&E staining (Sigma-Aldrich, St. Louis, MO, USA). Sections were visualized under a BX51 microscope (Olympus, Tokyo, Japan). For immunohistochemical staining, epididymal fat pad samples (*n* = 3 mice/group) were embedded in paraffin and cut into 5-mm sections. Deparaffinized sections of adipose tissue were placed in 0.3% H_2_O_2_ for 30 min, washed, and incubated in blocking serum for 1 h at room temperature. Sections were incubated in primary antibodies (Appendix A) at 4 °C overnight and with a secondary biotinylated antibody for 1 h at room temperature. After washing, sections were incubated in an avidin–biotin–peroxidase complex solution (Vector Laboratories) and developed with a 0.05% diaminobenzidine substrate kit (Vector Laboratories). For double immunofluorescence, sections of deparaffinized epididymal fat tissue were incubated at 4 °C overnight with the primary antibodies (Appendix A). Double immunofluorescence procedures were performed as previously described [38]. Additionally, we performed double immunofluorescence staining for SHIP1 and TUNEL to measure the degree of SHIP1-positive apoptotic adipocytes using an in situ cell death detection kit (Roche Molecular Biochemicals, Mannheim, Germany) according to the manufacturer’s instructions. Nuclei were counterstained with DAPI (Invitrogen, Carlsbad, CA, USA). Sections were visualized with a microscope (BX51, Olympus, Tokyo, Japan), and digital images were captured.

### 4.6. Protein Extractions and Western Blot Analyses

Total lysates were prepared from adipose tissue in epididymal fat (*n* = 3–6 per group), as previously described [39]. Protein concentrations were determined using a BioRad protein assay, and samples were stored at −80 °C until use. Western blot analyses were performed using standard methods. Membranes were probed with each primary antibody (Appendix A). The Multi-Gauge image analysis program (version 3.0; Fujifilm, Tokyo, Japan) was used for quantification. To normalize the protein levels, α-tubulin was used as the internal control.

### 4.7. Statistical Analyses

Differences between groups were assessed using one-way analysis of variance (ANOVA), followed by post hoc analysis with Tukey’s test (Gaphpad Prism 7.0, GraphPad Software Inc., San Diego, CA, USA). A *p*-value less than 0.05 was considered significant. Results are expressed as the means ± standard of error of the mean (SEM).

## Figures and Tables

**Figure 1 ijms-21-07225-f001:**
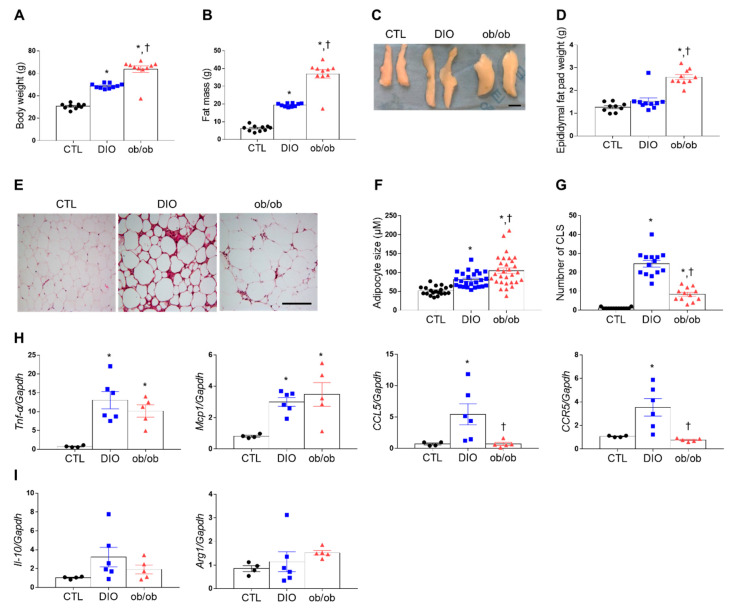
Effects of high-fat diet and leptin deficiency on weight gain, fat weight, and adiposity. (**A**) Body weight of mice after 20 weeks of feeding (*n* = 10 per group). (**B**) Fat mass weight measured by Echo Magnetic Resonance Imaging (EchoMRI). (**C**,**D**) Representative photographs (**C**) and weight (**D**) of epididymal adipose tissue. Scale bar = 1 cm. (**E**) Representative images (200×) of hematoxylin and eosin (H&E) staining in adipose tissue sections. Scale bar = 100 μm. (**F**) Adipocyte size (*n* = 3). (**G**) Number of crown-like structures (CLSs) measured by H&E-stained epididymal fat (*n* = 3). (**H**,**I**) Quantitative RT-PCR analysis of tumor necrosis factor-α (Tnf-α), monocyte chemoattractant protein 1 (Mcp1), CC chemokine ligand 5 (CCL5), C-C motif receptor 5 (CCR5), interleukin-10 (Il-10), and arginase 1 (Arg1) mRNA. Data are presented as the means ± standard of error of the mean (SEM). * *p* < 0.05 versus control (CTL) mice; † *p* < 0.05 versus diet-induced obesity (DIO) mice. CTL (black circle), DIO (blue square), ob/ob (red triangle).

**Figure 2 ijms-21-07225-f002:**
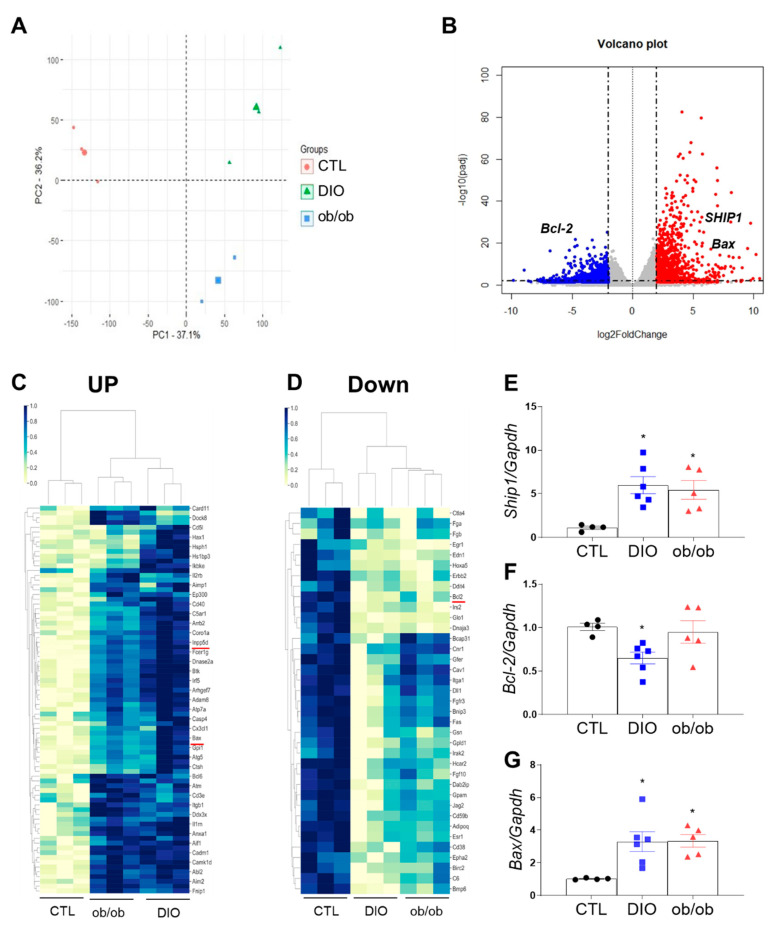
RNA-seq analysis of adipose tissue. (**A**) Principal component analysis for 9 samples from CTL, DIO, and ob/ob mice (3 replicates for each group). (**B**) Volcano plot shows the differential gene expressions as blue and red dots (|fold change| < 1.60). (**C**,**D**) Cluster map of differential gene expression related to immune system and apoptosis showing upregulated (**C**) and downregulated genes (**D**) in CTL mice relative to DIO or ob/ob mice. (**E**–**G**) Quantitative RT-PCR analysis of Src homology domain-containing inositol 5′-phosphatase 1 (Ship1) (**E**), Bcl-2 (**F**), and Bax (**G**). Data are presented as the means ± SEM; * *p* < 0.05 versus CTL mice. CTL (black circle), DIO (blue square), ob/ob (red triangle).

**Figure 3 ijms-21-07225-f003:**
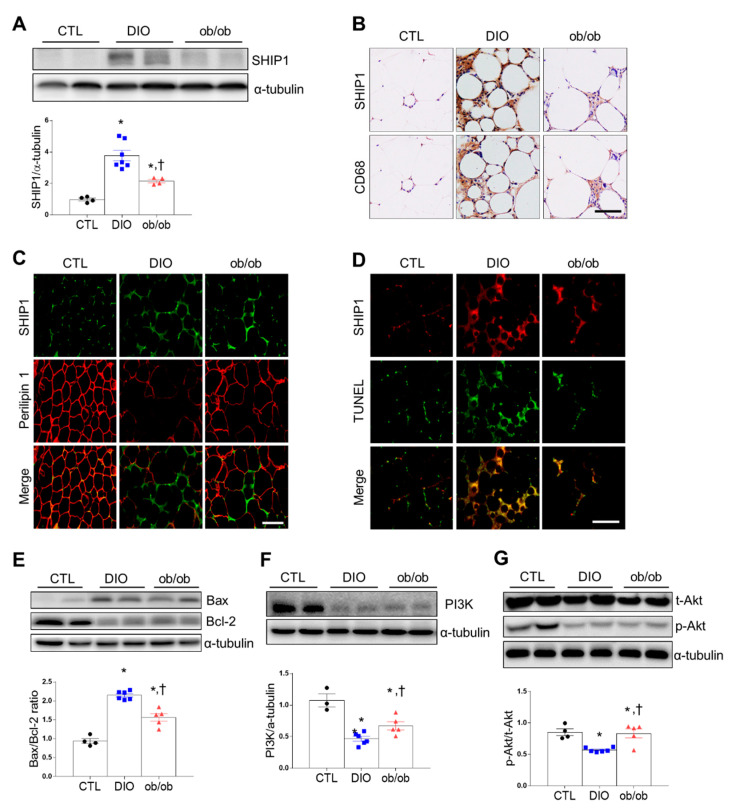
Effects of high-fat diet and leptin deficiency on inflammation and apoptosis in adipose tissue. (**A**) Western blot analysis of SHIP1 protein expression. (**B**) Representative immunostained SHIP1 and Cluster of Differentiation 68 (CD68) in serial adipose tissue sections (200×). (**C**) Representative immunofluorescent images (200×) of SHIP1 (green) and perilipin 1 (red). (**D**) Representative immunofluorescent images (200×) of SHIP1 (red) and terminal deoxynucleotidyl transferase dUTP nick-end labeling (TUNEL) (green). (**E**) Western blot analysis of Bax and Bcl-2 protein. Graph indicates the Bax/Bcl-2 ratio. (**F**,**G**) Western blot analysis of phosphoinositide 3-kinase (PI3K),total protein kinase B (t-Akt), and phosphorylated Akt (p-Akt) protein expression, using α-tubulin as a reference protein. Data are presented as the means ± SEM. * *p* < 0.05 versus CTL mice; † *p* < 0.05 versus DIO mice. Scale bar = 50 μm. CTL (black circle), DIO (blue square), ob/ob (red triangle).

**Figure 4 ijms-21-07225-f004:**
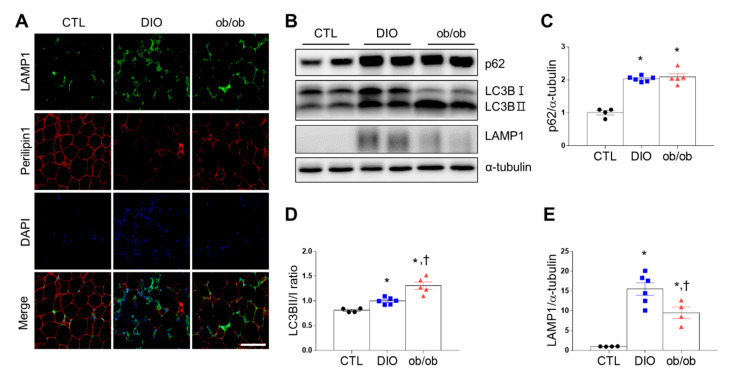
Effects of high-fat diet and leptin deficiency on autophagy in adipose tissue. (**A**) Representative immunofluorescent images (200×) of lysosomal-associated membrane protein 1 (LAMP1) (green), perilipin 1 (red), and 4′,6-diamidino-2-phenylindole (DAPI) (blue) in adipose tissue. (**B**) Western blot analysis of p62, LC3B, and LAMP1 protein expression, using α-tubulin as a reference protein. (**C**–**E**) Quantitative Western blot analysis of p62 (**C**), LC3BII/I ratio (**D**), and LAMP1 (**E**) in adipose tissue. Data are presented as the means ± SEM. * *p* < 0.05 versus control (CTL) mice; † *p* < 0.05 versus diet-induced obesity (DIO) mice. Scale bar = 100 μm. CTL (black circle), DIO (blue square), ob/ob (red triangle).

**Figure 5 ijms-21-07225-f005:**
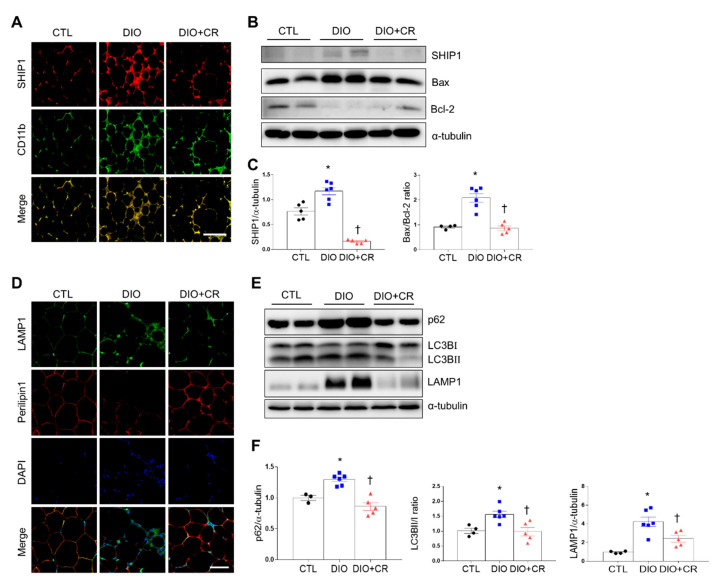
Effects of caloric restriction (CR) on SHIP1 expression in the adipose tissue of diet-induced obesity (DIO) mice. (**A**) Representative immunofluorescent images (200×) of SHIP1 (red) and CD11b (green) in the adipose tissue of CTL mice relative to DIO mice or DIO + CR mice. (**B**) Western blot analysis of SHIP1, Bax, and Bcl-2 protein expression, using α-tubulin as a reference protein. (**C**) Quantitative Western blot analysis of SHIP1 and Bax/Bcl-2 ratio in adipose tissue. (**D**) Representative immunofluorescent images (200×) of LAMP1 (green) and perilipin 1 (red) in the adipose tissue of CTL mice relative to DIO mice or DIO + CR mice. (**E**) Western blot analysis of p62, LC3B, and LAMP1 protein expression, using α-tubulin as a reference protein. (**F**) Quantitative Western blot analysis of p62, LC3BII/I ratio, and LAMP1 in adipose tissue. Data are presented as the means ± SEM; * *p* < 0.05 versus CTL mice; † *p* < 0.05 versus DIO mice. Scale bar = 50 μm. CTL (black circle), DIO (blue square), DIO+CR (red triangle).

**Figure 6 ijms-21-07225-f006:**
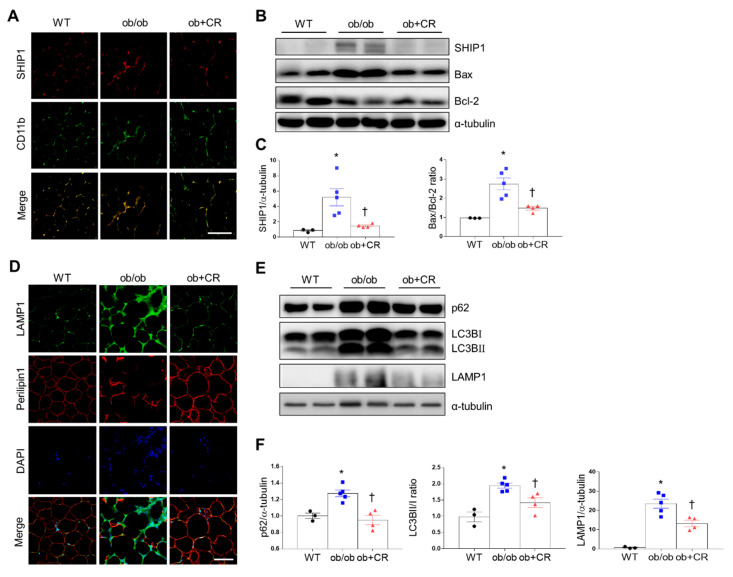
Effects of caloric restriction (CR) on SHIP1 expression in the adipose tissue of *ob/ob* mice. (**A**) Representative immunofluorescent images (200×) of SHIP1 (red) and CD11b (green) in the adipose tissue of control (CTL) mice relative to ob/ob mice or ob + CR mice. (**B**) Western blot analysis of SHIP1, Bax, and Bcl-2 protein expression, using α-tubulin as a reference protein. (**C**) Quantitative Western blot analysis of SHIP1 and Bax/Bcl-2 ratio in adipose tissue. (**D**) Representative immunofluorescent images (200×) of LAMP1 (green) and perilipin 1 (red) in the adipose tissue of CTL mice relative to ob/ob mice or ob + CR mice. (**E**) Western blot analysis of p62, LC3B, and LAMP1 protein expression, using α-tubulin as a reference protein. (**F**) Quantitative Western blot analysis of p62, LC3BII/I ratio, and LAMP1 in adipose tissue. Data are presented as the means ± SEM; * *p* < 0.05 versus CTL mice; † *p* < 0.05 versus ob/ob mice. Scale bar = 50 μm. WT, wild type. CTL (black circle), ob/ob (blue square), ob/ob+CR (red triangle).

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
