# Peer review of "The Role of SHIP1 on Apoptosis and Autophagy in the Adipose Tissue of Obese Mice"

_ijms, 2020, doi:10.3390/ijms21197225_

Round 1
Reviewer 1 Report
I accept the manuscript in the present form.
Check lines 71-72
Reviewer 2 Report
The paper has improved significantly from the previous version, and I have no further comments to add. I believe that responses to reviewers' comments have significantly improved the quality of the research.
This manuscript is a resubmission of an earlier submission. The following is a list of the peer review reports and author responses from that submission.
Round 1
Reviewer 1 Report
It is a great and interesting research, with clear and interesting objectives and results. The research is well designed, and the paper is easy to read. The topic is of great interest, and I have only few comments about your paper.
Introduction:
1.- More information is lacking on the role of macrophages, specifically the M1 and M2 subpopulations, in the pathology of obesity.
Results:
2.- Fig 1A is not necessary for the paper (in my opinion). In addition, lines 65-66 (page 2) are also unnecessary.
3.- Why do you use gapdh as housekeeping gene? Did you only use one gene to normalize data? Did you check that gapdh is not affected by treatment or leptin defficiency?
4.- Body weight was measured in 10 mice, but other determinations were only performed in n=3 mice. How this affected data? What is the reason for such a significant reduction in sample size? For gene expression and western-blot data, may I suppose that all experiments were performed in n=3? In this case, what was the selection criteria? Was it randomized? I think this need further clarification.
5.- An interesting result and not discussed was the greater perilpin1 expression in control and ob-ob- mice compared to DIO. Do you know any previous reference in this regard?
6.- Lines 134-135, are you confirming that sentence or is just speculative? When you refer to obese individuals are you meaning obese mice?
7.- Figure 6 shows data about ob-ob- mice, but there are no data regarding DIO mice. Is there any reason for this?
Discussion
8.- Higher levels of autophagy in obese mice was observed (lines 193-194), but again, are you referring to leptin defficient mice or obese mice. Because there is a great difference, as you really know. In my opinion, this issue should be clarified in the discussion.
9.- I do not fully understand lines 196-197 (page 8)
10.- Several differences regarding gene expression in DIO and ob-ob- mice were observed. In lines 202-203, leptin levels were proposed as possible cause. Did you measure leptin levels in DIO or control mice?
11.- Line 227, you commented that SHIP1 also regulates adipogenesis. It seems a bit contradictory to your observations. For example, the activation of SHIP1, could promote adipogenesis or apoptosis? In my opinion, this is of great interest and should be further discussed. Similarly, the higher level of perilipin-1 free adipocytes in DIO mice (line 230) is of great interest and should be discussed in more detail.
MATERIALS AND METHODS
12.- An a priori analysis for sample size would be of great interest. Having such a few sample size limits the use of one-way ANOVA. Alternatively, I recommend you to employ non-parametric analyses. Other way, a post-hoc power analysis would be of great interest.
To sum up, the paper if very interesting, and many data of great interest are shown. A few grammatical errors should also be corrected.
Reviewer 2 Report
There are several concerns that should be addressed.
1- in paragraph 2.1 authors referred to macrophage infiltration, but they only performed a RT-PCR analysis of Mcp1.
2. what is the adipose tissue used for RNA extraction? visceral or subcutaneous one?
3. How authors isolated macrophages for western blot analysis?
4. instead of BAX/Bcl-2 ratio evaluation, I suggest the determination of caspase 3 activation.
4. How do explain AKT phoshorylation, if DIO and ob/ob mice are PI3K free?
Conclusion is not clearly supported by data shown.